# Antiseizure Effects of Scoparone, Borneol and Their Impact on the Anticonvulsant Potency of Four Classic Antiseizure Medications in the Mouse MES Model—An Isobolographic Transformation

**DOI:** 10.3390/ijms24021395

**Published:** 2023-01-11

**Authors:** Jarogniew J. Łuszczki, Hubert Bojar, Agnieszka Góralczyk, Krystyna Skalicka-Woźniak

**Affiliations:** 1Department of Occupational Medicine, Medical University of Lublin, 20-090 Lublin, Poland; 2Department of Toxicology and Food Safety, Institute of Rural Health, 20-090 Lublin, Poland; 3Department of Natural Products Chemistry, Medical University of Lublin, 20-093 Lublin, Poland

**Keywords:** borneol, scoparone, coumarin, antiseizure medication, isobolographic transformation, maximal electroshock

## Abstract

Numerous botanical drugs containing coumarins and terpenes are used in ethnomedicine all over the world for their various therapeutic properties, especially those affecting the CNS system. The treatment of epilepsy is based on antiseizure medications (ASMs), although novel strategies using naturally occurring substances with confirmed antiseizure properties are being developed nowadays. The aim of this study was to determine the anticonvulsant profiles of scoparone (a simple coumarin) and borneol (a bicyclic monoterpenoid) when administered separately and in combination, as well as their impact on the antiseizure effects of four classic ASMs (carbamazepine, phenytoin, phenobarbital and valproate) in the mouse model of maximal electroshock-induced (MES) tonic-clonic seizures. MES-induced seizures were evoked in mice receiving the respective doses of the tested natural compounds and classic ASMs (when applied alone or in combinations). Interactions for two-drug and three-drug mixtures were assessed by means of isobolographic transformation of data. Polygonograms were used to illustrate the types of interactions occurring among drugs. The total brain content of ASMs was measured in mice receiving the respective drug treatments with fluorescent polarization immunoassay. Scoparone and borneol, when administered alone, exerted anticonvulsant properties in the mouse MES model. The two-drug mixtures of scoparone with valproate, borneol with phenobarbital and borneol with valproate produced synergistic interactions in the mouse MES model, while the remaining tested two-drug mixtures produced additivity. The three-drug mixtures of scoparone + borneol with valproate and phenobarbital produced synergistic interactions in the mouse MES model. Verification of total brain concentrations of valproate and phenobarbital revealed that borneol elevated the total brain concentrations of both ASMs, while scoparone did not affect the brain content of these ASMs in mice. The synergistic interaction of scoparone with valproate observed in the mouse MES model is pharmacodynamic in nature. Borneol elevated the brain concentrations of the tested ASMs, contributing to the pharmacokinetic nature of the observed synergistic interactions with valproate and phenobarbital in the mouse MES model.

## 1. Introduction

Epilepsy is a chronic neurological disorder characterized by recurrent epileptic seizures. This condition is the fourth most common neurological disorder and affects approx. 1% of the world’s population, 80% of whom live in low- and middle-income countries [1]. To treat this disease, patients with epilepsy use antiseizure medications (ASMs) and may benefit from other therapies such as brain surgery or neuromodulation [2]. Despite these possibilities, a third of patients with epilepsy suffer from pharmacoresistance and side effects. There is, therefore, an urgent need for new, effective and affordable treatments to manage epilepsy [3]. 

Herbs used in traditional medicine remain the first-line treatment for most people with little or no access to ASMs [1,4]. The study of antiseizure effects of these plants using innovative in vivo assays and the identification of their bioactive principles are key aspects of providing pharmacological evidence for their use. This may also guide research on the discovery of compounds with original bioactivity profiles to solve some of the problems associated with current ASMs [5,6].

Numerous botanical drugs containing furanocoumarins are used in ethnomedicine all over the world for their stomachic, spasmolytic and sedative effects [7,8]. Previously, it has been documented that various naturally occurring coumarins exerted anticonvulsant properties in the mouse maximal electroshock-induced seizure (MES) model and potentiated the anticonvulsant potencies of classic ASMs. Of note, the MES test in rodents reflects tonic-clonic seizures and, to a certain extent, partial convulsions with or without secondary generalization in humans [9]. This seizure model is widely used in preclinical testing of substances with anticonvulsant properties [9]. Experimental studies revealed that imperatorin and osthole (two naturally occurring coumarins) produced anticonvulsant effects by themselves in the mouse MES model in time- and dose-dependent manners [10]. Additionally, the anticonvulsant screen test revealed that xanthotoxin, but not bergapten or oxypeucedanin, exerted anticonvulsant effects in the mouse MES model [11]. It has been found that coumarins potentiate the anticonvulsant effects of classic ASMs. More specifically, umbelliferone significantly potentiated the anticonvulsant action of phenobarbital (PB) and valproate (VPA) but not that of phenytoin (PHT) and carbamazepine (CBZ) in the mouse MES model [12]. Imperatorin potentiated the antiseizure potencies of CBZ, PHT and PB but not that of VPA [13]. Xanthotoxin enhanced the anticonvulsant action of CBZ and VPA but not that of PHT and PB in the mouse MES model [14]. In contrast, osthole had no significant impact on the antiseizure potencies of CBZ, PHT, PB and VPA in experimental animals subjected to MES-induced seizures [15,16]. Of note, the anticonvulsant effects of ASMs in the mouse MES model are usually expressed as the median effective doses (ED_50_—i.e., doses of the ASMs that protected 50% of the tested animals against tonic-clonic seizures). The mentioned coumarins potentiated the antiseizure activity of some classic ASMs when combined together by reducing their ED_50_ values. In other words, coumarins potentiated the anticonvulsant potency of some selected ASMs and less drug doses were needed to produce the same effect, i.e., a 50% protection from tonic-clonic seizures in the mouse MES model. Thus, a significant reduction in the ED_50_ values for ASMs observed after coumarin administration, as compared to the ASMs when administered alone, testifies the potentiation of the antiseizure effects by the coadministered coumarins.

Considering the abovementioned facts related to coumarins, we intend to determine the anticonvulsant profile of scoparone—another naturally occurring coumarin (Figure 1), whose molecular structure is closely related to umbelliferone—a potent coumarin with confirmed anticonvulsant properties [12]. Coumarins, whose structural core comprises a 2*H*-1-benzopyran-2-one scaffold, are an important group of organic compounds from natural and synthetic sources with actual and potential pharmaceutical relevance. Since functional group substitution can occur naturally or be introduced synthetically at each of the accessible sites in this scaffold, coumarins have evolved as a potentially interesting and still unexplored molecular framework for drug discovery [17,18]. In our previous research, for the first time, we evaluated the procognitive and anxiolytic effects of scoparone in mice as well as showed the potent effects of scoparone on lipid remodeling in the endocannabinoid system [19].

The anticonvulsant screen test in our earlier study revealed that borneol (a bicyclic monoterpene—Figure 1) also produced anticonvulsant properties in the mouse MES model [20]. Borneol, which, according to the theories of traditional Chinese medicine (TCM), is called an “upper guiding drug”, can direct drugs upward to the head, targeting the brain [21]. Recent studies also demonstrated that borneol assisted the permeation of drugs across the BBB and enhanced their distribution in brain tissue [21,22,23]. Given that coumarins present in plant-origin food can occur in combination with the BBB permeability enhancer borneol (e.g., in botanical drugs and TCM preparations), the effects of both naturally occurring compounds should be examined. It should be noticed that there are already well-established TCM pure natural preparations derived from plants existing and popularly used and marketed in China, where borneol is coadministered with coumarin-rich plants from the *Apiaceae* family. In the Huatuo Zaizao pill, for example, listed in the pharmacopeia of the People’s Republic of China and used to promote rehabilitation after stroke as a promoter of neurogenesis, borneol is included with five other Chinese herbs, among which is *Angelica sinensis* [24]. Borneol, as one of the compounds present in essential oils distilled from fruits of *Apiaceae* plants, already coexists naturally with coumarins, also widely present in fruits [25]. 

The aim of this study was to determine the anticonvulsant properties of scoparone and borneol in the mouse MES model as well as to assess their impact (when administered separately or in combination) on the anticonvulsant potencies of four classic ASMs (CBZ, PHT, PB and VPA) in the mouse MES model. 

## 2. Results

### 2.1. Time Course of the Anticonvulsant Effects of Scoparone and Borneol in the Mouse MES Model

Scoparone and borneol, when administered separately, exerted anticonvulsant effects in the mouse MES model. The time course of the antiseizure effects of both naturally occurring compounds revealed that scoparone and borneol produced their maximal antiseizure effects 15 min. after drug administration (Figure 2A,C). The antiseizure effects of scoparone and borneol were observed up to 120 min. after drug systemic (i.p.) administration (Figure 2A,C). The experimentally derived ED_50_ values for scoparone ranged from 199.8 mg/kg to 320.1 mg/kg (Figure 2B), whereas those for borneol ranged from 255.4 mg/kg to 448.1 mg/kg (Figure 2D). In both cases, statistical analysis revealed the existence of a linear trend between doses of the studied compounds and pretreatment times (Figure 2). The linear trend indicates that scoparone and borneol exerted anticonvulsant effects in time- and dose-dependent manners.

### 2.2. Effects of Scoparone and Borneol on the Anticonvulsant Potency of Four Classic Antiseizure Medications in the Mouse MES Model

Scoparone administered systemically at a fixed dose of 50 mg/kg had no impact on the antiseizure effects of CBZ and PHT in the mouse MES model (Table 1). In contrast, scoparone (50 mg/kg, i.p.) significantly enhanced the antiseizure potency of PB and VPA in the mouse MES model (Table 1). In the case of PB, scoparone reduced (by 35%) the ED_50_ value of the drug from 28.85 mg/kg to 18.71 mg/kg (** *p* < 0.01; Table 1). As regards VPA, scoparone diminished the ED_50_ value of VPA by 25% from 292.0 mg/kg to 219.4 mg/kg (*** *p* < 0.001; Table 1). Scoparone at a dose of 25 mg/kg had no impact on the antiseizure potential of CBZ, PHT and PB in the mouse MES model (Table 1), but it still significantly reduced the ED_50_ of VPA by 17% (* *p* < 0.05; Table 1). The lowest tested dose of scoparone (12.5 mg/kg) did not enhance the anticonvulsant action of VPA in the mouse MES model (Table 1). Borneol administered i.p. at a fixed dose of 50 mg/kg had no impact on the antiseizure effects of CBZ and PHT in the mouse MES model (Table 1). In contrast, borneol (50 mg/kg, i.p.) significantly enhanced the antiseizure potency of PB in the mouse MES model by reducing (by 32%) the ED_50_ value of the drug from 28.85 mg/kg to 19.57 mg/kg (* *p* < 0.05; Table 1). Similarly, borneol potentiated the antiseizure effect of VPA by diminishing its ED_50_ value by 35% from 292.0 mg/kg to 188.3 mg/kg (*** *p* < 0.001; Table 1). Borneol at a dose of 25 mg/kg had no impact on the antiseizure potential of CBZ, PHT and PB in the mouse MES model (Table 1), but it still considerably reduced the ED_50_ of VPA by 30% (** *p* < 0.01; Table 1). The lowest tested dose of borneol (12.5 mg/kg) did not enhance the anticonvulsant action of VPA in the mouse MES model (Table 1). 

### 2.3. Isobolographic Analysis of Two-Drug Interaction between Scoparone, Borneol and Four Classic Antiseizure Medications in the Mouse MES Model

The effective doses (ED_50_) of classic ASMs along with constant doses of scoparone and borneol were transformed isobolographically to precisely characterize the types of interactions for the studied combinations. With isobolography, it was possible to calculate the sum of fractions of the studied drugs in combination (Table 2). Statistical comparison of the experimentally derived ED_50_ values for the studied combinations of ASMs with scoparone (50 mg/kg) with their respective theoretically calculated additive ED_50_ values provided evidence that there was no significant difference between these values, confirming the additive nature of interactions (Table 2; Figure 3). Thus, with isobolographic transformation, it was found that the potentiation of the antiseizure activity of PB and VPA by scoparone (50 mg/kg) produced additive interactions in the mouse MES model (Table 2; Figure 3). 

Statistical analysis of the ED_50exp_ values for the studied combinations of ASMs with borneol (50 mg/kg) with their respective theoretically calculated additive ED_50add_ values provided evidence that there was no significant difference between these values, confirming the additive nature of interactions between borneol and CBZ, PHT and PB (Table 2; Figure 4). In contrast, the combination of borneol (50 mg/kg) with VPA produced a synergistic interaction in the mouse MES model because, for this combination, the ED_50exp_ value significantly differed from the ED_50add_ (** *p* < 0.01; Table 2; Figure 4). Thus, the potentiation of the antiseizure activity of VPA by borneol produced a supra-additive (synergistic) interaction in the mouse MES model. 

### 2.4. Isobolographic Analysis of Interaction for Three-Drug Mixture Containing Scoparone, Borneol and Four Classic Antiseizure Medications in the Mouse MES Model

The mixture of constant doses of scoparone (25 mg/kg) and borneol (25 mg/kg) was combined with CBZ, PHT, PB and VPA in the mouse MES model. Isobolographic analysis of data for the three-drug combinations, i.e., a classic ASM + scoparone (25 mg/kg) + borneol (25 mg/kg), revealed that the combinations of CBZ and PHT with the mixture of scoparone and borneol produced additive interactions in the mouse MES model (Table 3). In these cases, the ED_50exp_ values did not significantly differ from the corresponding ED_50add_ values (Table 3). On the contrary, the combination of PB and VPA with scoparone (25 mg/kg) + borneol (25 mg/kg) produced synergistic interactions in the mouse MES model because the experimentally derived ED_50exp_ values considerably differed from the additively calculated ED_50add_ values, at * *p* < 0.05 and *** *p* < 0.001, respectively (Table 3).

To illustrate the interactions for two-drug and three-drug mixtures in the mouse MES model, a polygonogram was drawn (Figure 5). This graphical presentation of interactions between drugs allows for a fast selection of the most beneficial combinations of the tested drugs in mixtures. In this study, we found that the most beneficial combination for two-drug mixtures was that offering synergy between borneol and VPA in the mouse MES model (Figure 5a). Similarly, for the three-drug mixtures, the combination of scoparone + borneol with VPA was also synergistic in the mouse MES model (Figure 5b). Additionally, the combination of scoparone + borneol with PB occurred synergistically in the mouse MES model (Figure 5b), and thus, this combination is also worthy of being recommended for further studies. 

### 2.5. Measurement of Total Brain Concentrations of Phenobarbital and Valproate for the Three-Drug Mixture in Mice

The mixture of constant doses of scoparone (25 mg/kg) and borneol (25 mg/kg) significantly elevated total brain concentrations of PB (by 50%; ** *p* < 0.01) as compared to the control animals (Figure 6a). Similarly, borneol (25 mg/kg) coadministered with PB significantly elevated (by 39%, * *p* < 0.05) the total brain content of the studied ASM (Figure 6a). In contrast, scoparone (25 mg/kg) when combined with PB had no significant effect on the total brain content of this ASM in the mice (Figure 6a). As regards VPA, the mixture of scoparone (25 mg/kg) and borneol (25 mg/kg) significantly increased total brain concentrations of VPA (by 29%; *** *p* < 0.001) as compared to the control (treated with VPA alone) animals (Figure 6b). Similarly, borneol (25 mg/kg) with VPA significantly raised the total brain content of the latter drug (by 22%, ** *p* < 0.05; Figure 6a). In contrast, scoparone (25 mg/kg) when combined with VPA had no significant effect on the total brain content of this ASM in the mice (Figure 6b). 

## 3. Discussion

Results presented herein indicate clearly that scoparone and borneol produced anticonvulsant effects when administered alone. Evaluation of time-course effects of these natural compounds revealed that the peak of the maximum anticonvulsant effect was observed 15 min after both scoparone and borneol systemic (i.p.) administration. The antiseizure effects of scoparone and borneol were observed up to 120 min after drug administration, but these effects diminished along with the prolongation of the pretreatment times. The decrease in the anticonvulsant effects of scoparone and borneol was manifested by the elevation of ED_50_ values for these natural agents in the tonic-clonic seizure (MES) model along with the prolongation of pretreatment times. 

A comparison of the anticonvulsant potencies of scoparone with those of the other coumarins revealed that the most efficacious drug in terms of tonic-clonic seizure suppression was imperatorin (Table 4). Xanthotoxin can be arranged on the second, and scoparone on the third, anticonvulsant ranking place (Table 4). Osthole and borneol are drugs with low anticonvulsant potential. 

Evaluation of the antiseizure potential of borneol (50 mg/kg) revealed that the compound potentiated the antiseizure effects of PB and VPA but not that of CBZ or PHT in the mouse model of tonic-clonic seizures. Additionally, we have observed that scoparone (50 mg/kg) potentiated the antiseizure effect of VPA and PB but not that of CBZ and PHT in the mouse MES model. The observed effects are in line with the results presented earlier for other coumarins tested in the mouse MES model, i.e., osthole, xanthotoxin, imperatorin and umbelliferone [10,12,27]. Of note, umbelliferone potentiated the anticonvulsant action of PB and VPA but not that of CBZ and PHT in the mouse MES model [12]. Xanthotoxin potentiated VPA and CBZ but not that of PHT or PB [27]. In contrast, imperatorin potentiated the anticonvulsant action of CBZ, PHT and PB but not that of VPA in the mouse MES model [13]. Only osthole (as a simple coumarin) had no significant impact on the antiseizure effect of four classic ASMs in the mouse tonic-clonic seizure model [15,16]. Since scoparone has a molecular structure similar to umbelliferone (Figure 1), it can exert the same types of interactions when combined with classic ASMs, as umbelliferone did in the mouse MES model. This hypothesis has been confirmed experimentally in this study. Evaluation of the antiseizure effects evoked by the drugs in mixtures was performed in this study by means of the isobolographic transformation of interactions. This is the reason for calculating the fractions of the antiseizure effects produced by fixed doses of the tested naturally occurring compounds and fractions of the classic ASMs. Summation of fractions allowed for the precise characterization of interactions between the studied drugs (for more detailed information, see [28,29]). Due to the isobolographic transformation, we found that the observed potentiation of the antiseizure effects of PB by scoparone or borneol in the two-drug mixture was additive in the mouse MES model. In the case of VPA, the interaction of this ASM with scoparone exerted additivity, while with borneol, the interaction was synergistic in nature (Table 2). However, for the three-drug mixtures, the combinations of scoparone + borneol with PB or VPA exerted synergistic interactions in the mouse MES model. In contrast, mixtures of the tested naturally occurring substances (borneol + scoparone) with CBZ and PHT exerted additive interactions in the mouse MES model following the isobolographic transformation of data (Table 3). To visualize the types of interactions between naturally occurring compounds (borneol and scoparone) with four classic ASMs in the mouse MES model, a polygonogram was created for both two-drug and three-drug mixtures. Due to such a simple graphical presentation of interactions, it was possible to quickly assess the types of interactions and select the most effective ones in terms of seizure suppression (Figure 5) for further experiments.

Measurement of total brain concentrations of VPA and PB allowed the precise characterization of three-drug interactions between classic ASMs and scoparone and borneol. Since scoparone did not significantly affect total brain VPA and PB content, the observed interaction of the ASM with scoparone was pharmacodynamic in nature. In contrast, borneol significantly elevated total brain concentrations of VPA and PB in the experimental animals, and thus, the observed synergistic interaction in the mouse MES model had a pharmacokinetic component. Previously, we have reported that only estimation of ASMs in the central compartment (brain tissue) provides full information about the nature of interactions occurring between the studied drugs. Estimation of total brain concentrations of PB and VPA confirmed that borneol exclusively elevated the concentrations of both ASMs in experimental mice. In contrast, scoparone had no significant effect on total brain concentrations of PB and VPA in the brain tissue of experimental mice. Verification of total brain concentrations of the ASM was conducted only for those combinations, producing synergistic interactions in the mouse MES model. Considering the measurement of ASM concentrations in this study, it can be stated that the observed synergistic interaction in the mouse MES model can be readily explained by the presence of pharmacokinetic interactions finally resulting in elevation of PB and VPA in brain tissue. Because the combinations of scoparone + borneol with CBZ and PHT exerted additive interactions in the mouse MES model, we did not verify their concentrations in brain tissue. This was in agreement with the 3R rule regarding the reduction in animals used during experiments to the minimal necessary number (MNN) so as to reach statistical relevance. This was the reason that we did not measure the total brain content of CBZ and PHT in experimental animals after administering borneol and scoparone.

Since borneol elevated the total brain concentrations of VPA and PB, it can be entirely responsible for the observed interaction in the mice subjected to the MES test. More specifically, it was observed that a 40% reduction in the ED_50_ value of VPA for the combination of VPA + scoparone + borneol in the MES test was accompanied with a 29% elevation of the total brain level of VPA in the mice. In the case of PB, a 43% reduction in the ED_50_ value of the ASM for the combination of PB + scoparone + borneol in the MES test was associated with a 50% elevation of the total brain level of PB in mouse brain tissue. No doubts exist that borneol affects the permeability of the BBB, causing elevation in PB and VPA concentrations in the brain tissue of experimental animals. It is worth mentioning that scoparone, when administered alone, did not significantly alter the brain content of PB or VPA in the mice. Of note, verification of ASM concentrations in the central compartment and the site of action of ASMs (i.e., neuronal tissue as a target) have been claimed previously, since assessment of ASM concentrations in the blood/plasma of experimental animals is not adequate due to significant differences between the plasma and brain concentrations [30]. Only the central compartment (brain) adequately reflects the observed interaction in mice [31,32]. In this study, we tested a scientific hypothesis that borneol, due to its special properties, can affect coadministered drugs. This hypothesis was confirmed experimentally since only VPA and PB responded positively and borneol enhanced the antiseizure potency of VPA and PB. Other tested ASMs, including CBZ and PHT, did not exert synergistic interactions in the mouse MES model. Previously, it was experimentally confirmed that the maximal concentrations of scoparone in brain tissue were observed 15 min. after drug i.p. administration, and the brain scoparone content decreased due to its rapid elimination from the brain compartment [19]. This was the reason that both tested compounds (borneol and scoparone) were administered in this study 15 min. prior to the MES test and brain tissue sampling, when the naturally occurring substances were coadministered with ASMs. Our results indirectly confirmed the previous findings documenting that borneol increased the brain permeability for scoparone [19], which occurred in our study for PB and VPA (two ASMs). 

Borneol was able to open the BBB, and thus, it started to be considered as an ideal promoter for brain-targeting drug delivery [22]. It was already shown that borneol promotes the accumulation of other drugs in brain tissue and increases the brain bioavailability of drugs, however with different effects in the different brain regions. Experiments conducted on four brain regions, the cortex, hippocampus, hypothalamus and striatum, showed that borneol at a dose of 200 mg/kg produces a loose structure in the tight junction and void structure between the endothelial cell and mesangial cell [21]. A pharmacokinetics study demonstrated that delivery of borneol in the cortex was the highest and in the striatum the lowest, with deliveries of borneol in the hippocampus and hypothalamus in between the two. Borneol showed a tissue-specific BBB-opening effect, which was associated with its regulation of the ultrastructure of brain tissues and the decreasing expression of the multidrug resistance proteins Mdr1a, Mdr1b and Mrp1 [21]. In addition, borneol can significantly inhibit the activity of P-gp in brain microvascular endothelial cells by an NF-kB-signaling-mediated mechanism, as shown in a BBB in vitro model [33]. 

So far, there is little evidence that borneol can enhance the BBB permeability of coumarin derivatives. Luo et al. [34] investigated the effect of borneol enantiomers on the pharmacokinetics of osthole. When osthole (300 mg/kg) was coadministered orally with (+)-borneol, (-)-borneol or synthetic borneol (all three compounds at a constant dose of 400 mg/kg) to male rats, the bioavailability (AUC value) of osthole was significantly enhanced by 78.17, 167.79 and 104.71%, respectively, when compared with those in the osthole + vehicle-treated rats. The Cmax value of osthole was significantly promoted by 51.77, 271.29 and 92.64%, respectively. The T1/2 of osthole increased by 115.75, 259.13 and 378.59%, respectively. There were significant differences between the borneol enantiomers, with (−)-borneol having a stronger promotional effect on the pharmacokinetics of osthole than (+)-borneol [34]. The bioactivity of coumarins is very often limited due to poor bioavailability and low plasma concentrations, which result from the rapid elimination by the CYP3A4 enzyme in the liver. It is, therefore, important to highlight that borneol is an in vivo inducer of the CYP2B and CYP3A enzymes [35,36]. 

Previously, a quantitative estimation of scoparone concentrations in mouse brain tissue using an LC-MS/MS method at four pretreatment times (i.e., 15, 30, 60 and 120 min.) revealed that scoparone administered i.p. at doses of 5 and 12.5 mg/kg underwent a rapid elimination (first-order kinetics) from the brain tissue [19]. The combined administration of scoparone (12.5 mg/kg) with borneol (50 mg/kg) resulted in a three-fold elevation in scoparone content in the brain tissue of experimental animals, which was also accompanied by appearance of isofraxidin’s traces (the main metabolite of scoparone) [19]. Considering the fact that scoparone underwent a rapid elimination (first-order kinetics) from mouse brain tissue, the pretreatment time of 15 min. seems to be more appropriate to measure the ASMs’ concentrations in this study.

Of note, the acute administration of the drugs (scoparone, borneol and ASMs), while testing the two-drug and three-drug combinations, seems to be a limitation in this study in relation to the extrapolation of the results to clinical situations. In patients with epilepsy, the ASMs and other adjuvant therapies are prescribed for a long period of time and the patients take their medications chronically. We are fully aware of the fact that the results from the acute experiments cannot be directly transferred to clinical conditions, but they could shed more light on general mechanisms responsible for the observed interactions. Full translatability of the results to clinical conditions could be reached after further more detailed investigations with chronic administration of the drugs. 

Sometimes, administration of the second or third drug may produce some adverse effects, and substantial reduction in doses of these drugs can be necessary. Administration of low doses of scoparone and borneol in the mixture with classic ASMs was beneficial due to the lack of side effects related to application of both naturally occurring compounds in low (non-toxic) doses. Due to the isobolographic transformation of data, it was possible to determine the anticonvulsant action of classic ASMs in the mouse MES model. Although more advanced studies are recommended to elucidate the mechanisms responsible for synergistic interactions between borneol, scoparone and VPA or PB in experimental animals, this study has some limitations related to the lack of determination of scoparone, its main metabolite (isofraxidin) and borneol content in the brain tissue of the mice receiving the ASMs in combinations with borneol and scoparone. Mutual verification of total brain concentrations of all the studied drugs and tested compounds provides us with proof that interactions observed in animals are certainly correct, and no additional pharmacokinetic changes in the ASMs’ parameters could mask the observed interactions in experimental animals. It is highly likely that all the tested drugs in three-drug mixtures can mutually influence their own metabolism, and the total brain content of these compounds may differ.

In experimental epileptology, during evaluation of the anticonvulsant properties of the tested ASMs, general toxicity produced by the drugs applied at the anticonvulsant doses is usually and routinely screened in the chimney test, which detects disturbances in the locomotor activity of the tested animals with respect to motor coordination movement and balance during the spontaneous behavior of experimental animals [37]. In the chimney test, any potential impairment in motor coordination, resulting from the drug treatment, is detected [37]. Additionally, high toxic doses of drugs may sometimes evoke muscular flaccidity, which can be detected during evaluation of the skeletal muscular strength in the grip strength test in animals [38]. Of note, no general toxicity (i.e., significant impairment of motor coordination or muscular flaccidity in mice) was observed in this study, but specific toxicity ascribed to both naturally occurring substances (scoparone and borneol) in the two-drug and three-drug mixtures was not tested in this study.

Another fact needs explanation while translating the observed effects in mice to clinical conditions. Since scoparone was rapidly eliminated from brain tissue and borneol considerably elevated brain ASM content, it can be suggested that the combinations of borneol and scoparone with ASMs may possess short-term efficacy in the mouse MES model. This may raise a question of how to prevent the decrease in the anticonvulsant effects associated with the elimination of scoparone from brain tissue. On the other hand, the short-term efficacy of such combinations may be used clinically to temporarily increase the total brain content of ASMs when preventing seizures from occurring or terminating seizure attacks. These translational questions remain unanswered now, but in the future, it will be possible to correct the pharmacokinetic profile of scoparone and/or borneol, making the combinations with ASMs clinically usable. 

## 4. Materials and Methods

### 4.1. Experimental Animals

The total number of animals used in this study was 256 adult (8–9-week-old) male albino Swiss outbred mice. Laboratory conditions before and during the experiments were in strict accordance with the EU Directive 2010/63/EU for animal experiments and complied with the ARRIVE guidelines. Each experimental group comprises 8 randomly selected mice. Each mouse was used only once, and all experimental tests were conducted between 08:00 a.m. and 03:00 p.m. The experimental protocols and procedures described below were approved by the local Ethics Committee at the University of Life Sciences in Lublin (license no.: 23/2018 from 12 February 2018).

### 4.2. Drugs

All drugs—borneol, carbamazepine, phenytoin, phenobarbital and valproate—were purchased from Sigma-Aldrich (Poznań, Poland). Scoparone was isolated from *Artemisia umbelliformis* Lam. (Asteraceae), as reported previously [19]. All the studied drugs, except for valproate (i.e., borneol, scoparone, carbamazepine, phenobarbital and phenytoin), were suspended in a 1% aqueous solution of Tween 80 (Sigma-Aldrich, Poznań, Poland), while valproate was dissolved in distilled water. The ASMs were administered *intraperitoneally* (*ip*) as follows: phenytoin—120 min, phenobarbital—60 min and carbamazepine and valproate—30 min, before the experiments, as reported earlier [39,40]. Borneol and scoparone were injected i.p. at increasing time periods of 15, 30, 60 and 120 min. Of note, these pretreatment times for borneol and scoparone when used separately (i.e., 15, 30, 60 and 120 min.) were based on our previously published work in order to make the results for scoparone comparable to those for other coumarins (i.e., osthole, imperatorin and xanthotoxin) tested earlier [10,26]. For all the combinations with ASMs, both borneol and scoparone were administered i.p. 15 min. before the MES test and brain tissue sampling.

### 4.3. Electrically Evoked Tonic-Clonic Seizures in Animals

The seizure activity in mice was induced experimentally by an alternating current (50 Hz; 500 V; 25 mA) delivered via ear-clip electrodes. During experiments, increasing doses of the tested drugs were administered to animals and the mice were subjected to electrically evoked tonic-clonic seizures. The percentage of animals protected against tonic-clonic seizures along with the respective doses of the drugs was expressed as the median effective dose (ED_50_) of the tested drugs, according to the log-probit method [41]. The ASMs administered alone and their combinations with scoprone, borneol and a mixture of both natural compounds were tested for their ability to increase the number of animals protected from tonic hind limb extension after electric stimulation. Each ED_50_ value represented a dose of the ASM (in mg/kg) predicted to protect 50% of the mice tested against electrically induced extension of the animals’ hind limbs. In this study, carbamazepine (CBZ) and phenytoin (PHT) were administered at doses ranging between 6 and 16 mg/kg, phenobarbital (PB) at doses ranging between 15 and 35 mg/kg and valproate (VPA) at doses ranging between 150 and 350 mg/kg.

### 4.4. Isobolographic Transformation of Data

Calculation of fractions of the studied drugs (i.e., borneol, scoparone and classic ASMs) along with subsequent isobolographic transformation of data was conducted in strict accordance with the procedure as described earlier [42]. After summation of the fractions of the studied drugs in mixture (i.e., borneol, scoparone and the classic ASMs), the isobolographic characterization of interactions was possible, as reported elsewhere [28,29]. To graphically illustrate the studied interactions, polygonograms were drawn. On this graph, both naturally occurring compounds were connected with classic ASMs by straight lines, which characterize additive interaction. A synergistic interaction was marked in red. In this study, isobolographic analysis was based on the transformation of constant doses of scoparone and borneol into points illustrating interactions, whose nature was additive and synergistic in the mouse MES model. The isobologram for the constant dose of one drug (i.e., scoparone and/or borneol) has a parallel line to the Y-axis, reflecting this dose of the naturally occurring compound, whereas increasing doses of the second drug (ASM) allow the creation of the isobole. Such transformation takes into account doses of all drugs present in mixture. Calculation of fractions of the drugs in mixture contributes to the proper classification of interactions between the studied drugs, as reported earlier [28,29]. Interactions observed in the mouse MES model were presented graphically by means of polygonograms for two-drug and three-drug mixtures, as reported earlier [43,44] The polygonogram is the simplest and quickest way to illustrate synergistic interactions among the tested combinations, and this method of presentation is widely applied in isobolographic studies [45].

### 4.5. Measurement of Total Brain ASM Concentrations

The content of PB and VPA in the brain tissue of experimental animals was measured by fluorescence polarization immunoassay, as reported earlier [44,46]. Both PB and VPA were administered i.p. either alone or in combination with borneol and/or scoparone. The doses of PB and VPA reflected their ED_50_ values from the tonic-clonic seizure model. Preparation of brain tissue and measurement of ASM content have been described elsewhere [44,46]. Briefly, each mouse pretreated with the respective treatment (ASM alone, ASM with scoparone, ASM with borneol or ASM with scoparone and borneol) was decapitated at the time reflecting the peak of maximum anticonvulsant effect of the ASM in the MES test; subsequently, a whole mouse brain was removed from the skull, weighted, harvested and homogenized in the presence of Abbott buffer (1:2 weight/volume). The homogenates were centrifuged at 10.000 g for 10 min. and the supernatant was transferred to Abbott Architect c4000 analyzer (Abbott, Abbott Park, IL, USA) to perform fluorescence polarization immunoassay (FPIA) analysis with the manufacturer’s supplied kits for detection of PB and VPA. Total brain concentrations of ASMs were expressed in μg/g of wet brain tissue as means ± SD of 8 separate brain preparations.

### 4.6. Statistical Analysis

The ED_50_ values (± SEM) for borneol, scoparone and four classic ASMs were calculated from the log-probit method [41] and the method transforming 95% confidence limits to SEM, as described in detail earlier [47]. Statistical comparison of the respective ED_50_ values was performed with one-way ANOVA followed by the *post-hoc* Tukey–Kramer test. Statistical comparison of the respective ED_50exp_ and ED_50add_ values for each combination was performed with the unpaired Student’s t-test. The total brain content of ASMs for variations of mixtures of the tested drugs was analyzed with one-way ANOVA followed by Dunnett’s multiple comparisons test. Differences among ED_50_ values were considered statistically significant if *p* < 0.05.

## 5. Conclusions

Scoparaone and borneol exerted clear-cut anticonvulsant effects in time- and dose-dependent manners in animals subjected to the MES model. The synergistic interaction of scoparone with valproate observed in the mouse MES model for the two-drug mixture is pharmacodynamic in nature. The two-drug combinations of scoparone with carbamazepine, phenytoin and phenobarbital are additive in the mouse MES model. Borneol (administered either separately with classic ASMs or combined with scoparone), significantly elevated the brain concentrations of the tested ASMs, contributing to the pharmacokinetic nature of observed synergistic interactions with valproate and phenobarbital in the mouse MES model. Borneol in two-drug and three-drug mixtures with carbamazepine or phenytoin exerted additive interactions in the mouse MES model.

## Figures and Tables

**Figure 1 ijms-24-01395-f001:**
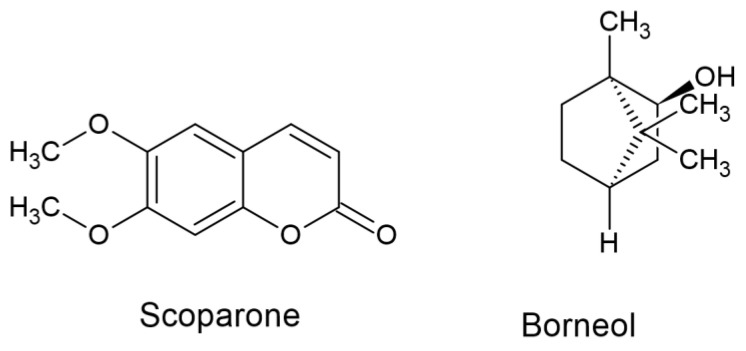
Chemical structure of scoparone and borneol.

**Figure 2 ijms-24-01395-f002:**
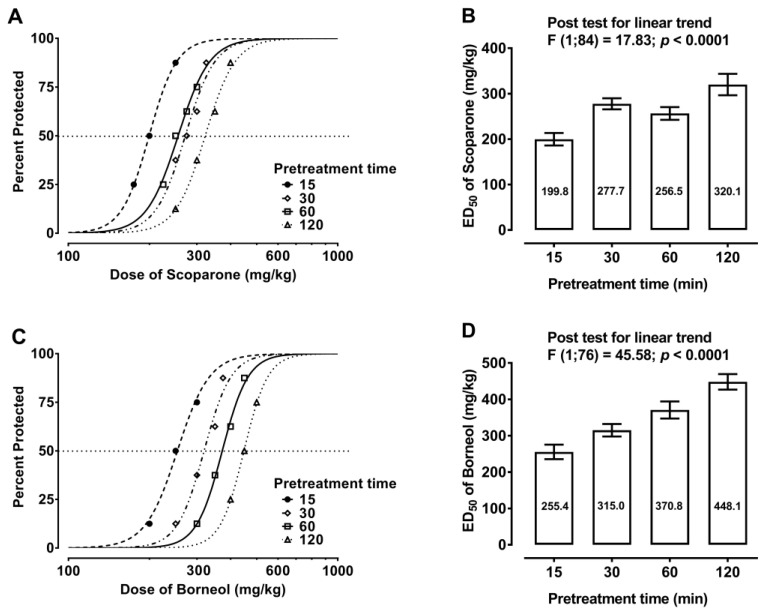
Time course of the anticonvulsant effects of scoparone and borneol in the mouse MES model. (**A**,**C**) Time course and dose–response relationship curves for scoparone and borneol administered systemically (i.p.) in the mouse MES model. The dotted line indicates the approx. ED_50_ values for scoparone and borneol, respectively. (**B**,**D**) Columns represent the ED_50_ values for scoparone and borneol when administered i.p. at various pretreatment times.

**Figure 3 ijms-24-01395-f003:**
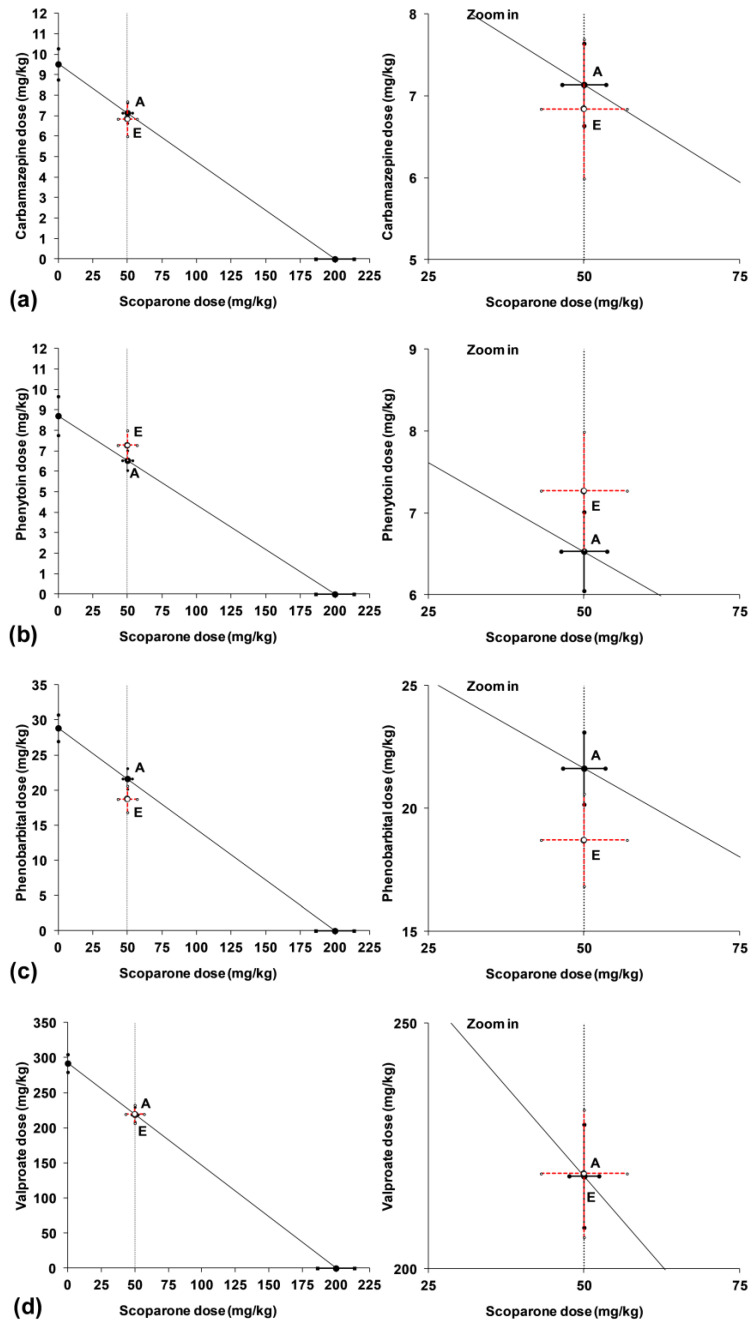
Isobolographic transformation of data for interactions between classic antiseizure medications and scoparone in the mouse MES model. Interactions between scoparone and carbamazepine (**a**), phenytoin (**b**), phenobarbital (**c**) and valproate (**d**) are plotted on isobolograms. The constant dose of scoparone in mixture is parallel to the Y-axis. The oblique line connecting the ED_50_ values of X- and Y-axes (as the line of additivity) reflects the theoretically calculated ED_50add_ values. Point A reflects the theoretically calculated ED_50add_ value for the two-drug mixture (scoparone + ASM), whereas the experimentally derived ED_50exp_ is illustrated graphically as point E.

**Figure 4 ijms-24-01395-f004:**
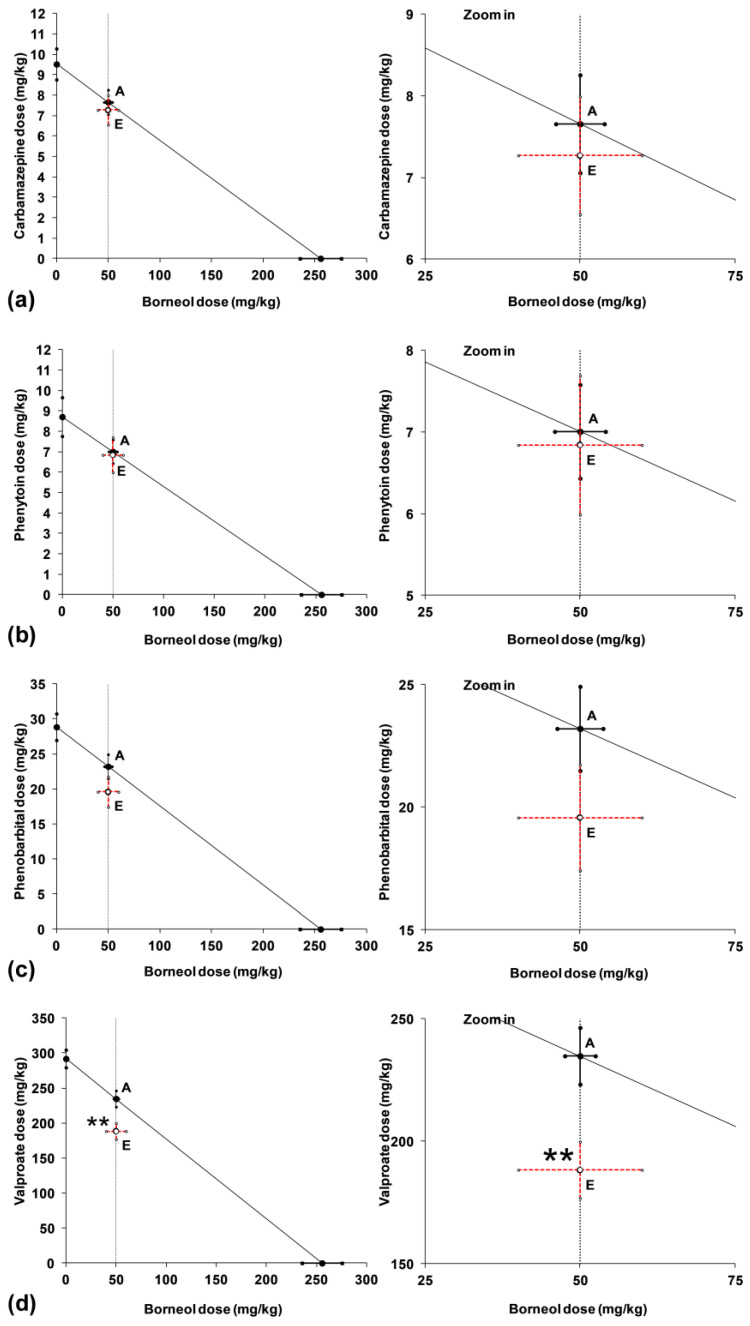
Isobolographic transformation of data for interactions between classic antiseizure medications and borneol in the mouse MES model. Interactions between borneol and carbamazepine (**a**), phenytoin (**b**), phenobarbital (**c**) and valproate (**d**) are plotted on isobolograms. The constant dose of borneol in mixture is parallel to the Y-axis. The oblique line connecting the ED_50_ values of X- and Y-axes (as the line of additivity) represents the theoretically calculated ED_50add_ values. Point A reflects the theoretically calculated ED_50add_ value for the two-drug mixture (borneol + ASM), whereas the experimentally derived ED_50exp_ is illustrated graphically as point E. ** *p* < 0.01 vs. the respective ED_50add_ value for valproate + borneol (point A).

**Figure 5 ijms-24-01395-f005:**
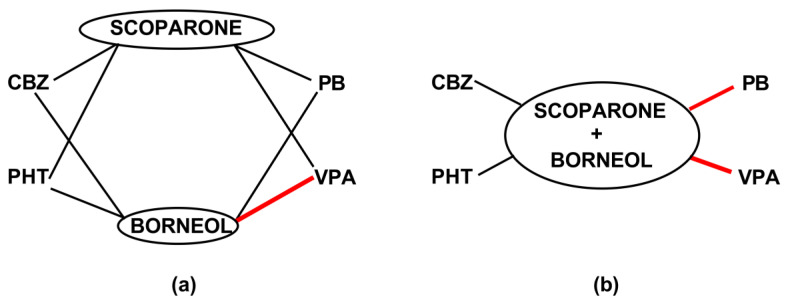
Polygonogram illustrating interactions for two-drug (**a**) and three-drug (**b**) mixtures among scoparone, borneol and four classic antiseizure medications in the mouse MES model. Black lines illustrate the additive interactions, whereas the red lines indicate synergistic interactions.

**Figure 6 ijms-24-01395-f006:**
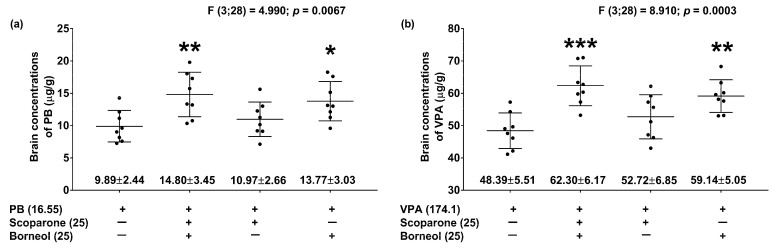
Influence of scoparone and borneol on total brain concentrations of PB (**a**) and VPA (**b**) in experimental animals. Scatter plots with whiskers illustrate mean concentrations (in μg/g of wet brain tissue ± SD of 8 determinations) of PB (**a**) and VPA (**b**) in the brain of experimental animals. Statistical analysis of data was performed with one-way ANOVA followed by Dunnett’s multiple comparisons test. Brain tissue samples were quantified using fluorescence polarization immunoassay. All the drugs were administered i.p. at doses corresponding to their ED_50_ values from the mouse MES model. PB—phenobarbital; VPA—valproate; * *p* < 0.05, ** *p* < 0.01 and *** *p* < 0.001 vs. the respective control (PB or VPA alone) animals.

**Table 1 ijms-24-01395-t001:** Influence of scoparone and borneol on the anticonvulsant potency of four classic antiseizure medications (ASMs) in the mouse MES model.

ASM	ED_50_	n	ANOVA Statistics	Fraction of ASM
CBZ + vehicle	9.52 ± 0.76	16		1.00
CBZ + scoparone (25)	8.44 ± 0.84	16		0.89
CBZ + scoparone (50)	6.84 ± 0.85	16	F(2;45) = 2.719; *p* = 0.077	0.72
PHT + vehicle	8.71 ± 0.95	16		1.00
PHT + scoparone (25)	8.18 ± 0.80	16		0.94
PHT + scoparone (50)	7.27 ± 0.72	16	F(2;45) = 0.772; *p* = 0.468	0.84
PB + vehicle	28.85 ± 1.89	16		1.00
PB + scoparone (25)	23.17 ± 2.18	16		0.80
PB + scoparone (50)	18.71 ± 1.88 **	16	F(2;45) = 6.534; *p* = 0.003	0.65
VPA + vehicle	292.0 ± 12.6	24		1.00
VPA + scoparone (12.5)	269.7 ± 12.9	24		0.92
VPA + scoparone (25)	241.5 ± 12.5 *	24		0.83
VPA + scoparone (50)	219.4 ± 13.0 ***	24	F(3;92) = 6.234; *p* = 0.0007	0.75
CBZ + vehicle	9.52 ± 0.76	16		1.00
CBZ + borneol (25)	7.83 ± 0.86	24		0.82
CBZ + borneol (50)	7.27 ± 0.72	16	F(2;53) = 1.772; *p* = 0.180	0.76
PHT + vehicle	8.71 ± 0.95	16		1.00
PHT + borneol (25)	7.82 ± 0.85	24		0.90
PHT + borneol (50)	6.84 ± 0.85	16	F(2;53) = 0.941; *p* = 0.397	0.79
PB + vehicle	28.85 ± 1.89	16		1.00
PB + borneol (25)	22.35 ± 2.09	24		0.78
PB + borneol (50)	19.57 ± 2.15 *	24	F(2;61) = 4.401; *p* = 0.016	0.68
VPA + vehicle	292.0 ± 12.6	24		1.00
VPA + borneol (12.5)	230.4 ± 11.3	16		0.79
VPA + borneol (25)	205.2 ± 11.7 **	16		0.70
VPA + borneol (50)	188.3 ± 11.5 ***	16	F(3;68) = 15.80; *p* < 0.0001	0.65

Results are presented as median effective dose (ED_50_) values (in mg/kg ± SEM) of the classic ASMs. n—number of animals at those doses whose anticonvulsant effects ranged between 4th and 6th probit. * *p* < 0.05, ** *p* < 0.01 and *** *p* < 0.001 vs. ASM + vehicle; ASM—antiseizure medication; CBZ—carbamazepine, PHT—phenytoin, PB—phenobarbital, VPA—valproate.

**Table 2 ijms-24-01395-t002:** Isobolographic transformation of the anticonvulsant potency of scoparone, borneol and four classic antiseizure medications in the mouse MES model.

Combination	ED_50exp_	n_exp_	ED_50add_	n_add_	*t*-Test Statistics	Sum of Fractions
CBZ + scoparone (50)	56.84 ± 0.85	16	57.14 ± 2.01	28	t = 0.138; df = 35.48; *p* = 0.891	0.72 + 0.25 = 0.97
PHT + scoparone (50)	57.27 ± 0.72	16	56.53 ± 2.08	28	t = 0.336; df = 33.01; *p* = 0.739	0.84 + 0.25 = 1.09
PB + scoparone (50)	68.71 ± 1.88	16	71.63 ± 2.44	28	t = 0.948; df = 41.96; *p* = 0.349	0.65 + 0.25 = 0.90
VPA + scoparone (50)	269.4 ± 12.96	24	268.9 ± 6.44	36	t = 0.035; df = 34.38; *p* = 0.973	0.75 + 0.25 = 1.00
CBZ + borneol (50)	57.27 ± 0.72	16	57.66 ± 2.26	28	t = 0.164; df = 32.16; *p* = 0.870	0.76 + 0.20 = 0.96
PHT + borneol (50)	56.84 ± 0.85	16	57.00 ± 2.33	28	t = 0.065; df = 33.59; *p* = 0.949	0.79 + 0.20 = 0.99
PB + borneol (50)	69.57 ± 2.15	24	73.20 ± 2.71	28	t = 1.049; df = 48.93; *p* = 0.299	0.68 + 0.20 = 0.88
VPA + borneol (50)	238.3 ± 11.48 **	16	284.8 ± 7.01	36	t = 3.457; df = 26.68; *p* = 0.002	0.65 + 0.20 = 0.85

Results are presented as ED_50exp_ and ED_50add_ values (in mg/kg ± SEM) of the classic ASMs in combination with scoparone, borneol or both drugs in the mixture. Both scoparone and borneol were administered i.p. at a constant dose of 50 mg/kg; n—number of animals at those doses whose anticonvulsant effects ranged between 4th and 6th probit. ** *p* < 0.01 vs. the respective ED_50add_ value.

**Table 3 ijms-24-01395-t003:** Isobolographic transformation of the anticonvulsant potency of three-drug combination—scoparone, borneol and four classic antiseizure medications in the mouse MES model.

Combination	ED_50exp_	n_exp_	ED_50add_	n_add_	*t*-Test Statistics	Sum of Fractions
CBZ + scoparone (25) + borneol (25)	56.40 ± 0.87	16	58.46 ± 2.26	28	t = 0.851; df = 34.24; *p* = 0.401	0.67 + 0.13 + 0.10 = 0.90
PHT + scoparone (25) + borneol (25)	55.21 ± 0.70	16	57.74 ± 2.34	28	t = 1.036; df = 31.59; *p* = 0.308	0.60 + 0.13 + 0.10 = 0.83
PB + scoparone (25) + borneol (25)	66.55 ± 2.40 *	16	75.64 ± 2.76	28	t = 2.485; df = 41.04; *p* = 0.017	0.57 + 0.13 + 0.10 = 0.80
VPA + scoparone (25) + borneol (25)	224.1 ± 11.01 ***	24	288.4 ± 7.51	36	t = 4.825; df = 43.23; *p* < 0.0001	0.60 + 0.13 + 0.10 = 0.83

Results are presented as ED_50exp_ and ED_50add_ values (in mg/kg ± SEM) of the classic ASMs in combination with scoparone and borneol. Both scoparone and borneol were administered i.p. at a constant dose of 25 mg/kg; n—number of animals at those doses whose anticonvulsant effects ranged between 4th and 6th probit. * *p* < 0.05 and *** *p* < 0.001 vs. the respective ED_50add_ value.

**Table 4 ijms-24-01395-t004:** Anticonvulsant potencies of various naturally occurring compounds in the mouse MES model.

Compound	Pretreatment Time (min)				Ref.
	15	30	60	120	
Imperatorin	185 (0.684)	167 (0.618)	206 (0.762)	290 (1.073)	[10]
Xanthotoxin	227.7 (0.910)	225.1 (0.899)	219.1 (0.875)	252.4 (1.008)	[26]
Scoparone	199.8 (0.969)	277.7 (1.347)	256.5 (1.244)	320.1 (1.552)	this study
Osthole	266 (1.089)	253 (1.036)	472 (1.932)	639 (2.616)	[10]
Borneol	255.4 (1.656)	315.0 (2.042)	370.8 (2.404)	448.1 (2.905)	this study

Results are presented as the ED_50_ values in mg/kg and (in mM in parentheses).

## Data Availability

Data are contained within the article.

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
