# Peer review of "Antiseizure Effects of Scoparone, Borneol and Their Impact on the Anticonvulsant Potency of Four Classic Antiseizure Medications in the Mouse MES Model—An Isobolographic Transformation"

_ijms, 2023, doi:10.3390/ijms24021395_

Round 1

Reviewer 1 Report

General comment:

The study by Łuszczki et al nicely explored the anticonvulsant profiles of the terpene derivative borneol and coumarin scoparone, either alone or in combination with widely-used anti-seizure medications, using a well-established generalized tonic-clonic seizure mouse model. The pairing of in vivo experimental data with isobologram to understand drug interaction is commendable. Considering that these two chemicals are widely used especially in traditional medicine, the findings are important as they contribute to the limited scientific evidence regarding anticonvulsant effect of naturally-occurring chemicals.

Specific comments:

1.       Duration of efficacy: Longer pre-treatment time for both borneol and scoparone mostly shifted the dose-response to the right (Fig 2), suggesting that the half-life for both is short. A study earlier this year by the same group (Kowalczyk et al 2022; Sci Rep 12, 822 (2022)) also reported that scoparone is rapidly eliminated from the brain, which could reduce the duration of efficacy. This raises a feasibility/translational issue and should be discussed.

2.       Toxicity: Side effects were a concern for ASMs but not mentioned for borneol and scoparone. The same study by Kowalczyk et al 2022 reported that low nanomolar concentrations of scoparone in the brain were sufficient to induce anxiogenic effect in mice, and that combination of scoparone (12.5 mg/kg) and borneol not only significantly increased brain scoparone concentration but also the level of the toxic compound isofraxidin. Doses more than 200 mg/kg were needed to achieve 50% protection from MES (Fig 2). In two-drug combination assays, 50 mg/kg dose was sometimes needed to achieve significant efficacy (Table 1), while 25 mg/kg was used to achieve efficacy in three-drug combination (Table 3). Authors should discuss the potential toxicity of using such high doses and how that might impact feasibility.

3.       Reference #41 (Łuszczki, J.J.; Czuczwar, S.J. 2003) cited in section 4.5 (line 451-458) did not report about fluorescence polarization immunoassay method or preparation of brain tissue/measurement of ASM content. Please amend reference and describe the details in methods including how and when the brains were collected.

4.   In table 2, it was assumed that the number 50 in bracket represents the dose. If so, please state in the table legend.

Author Response

REV_1

Specific comments:

  1. Duration of efficacy: Longer pre-treatment time for both borneol and scoparone mostly shifted the dose-response to the right (Fig 2), suggesting that the half-life for both is short. A study earlier this year by the same group (Kowalczyk et al 2022; Sci Rep 12, 822 (2022)) also reported that scoparone is rapidly eliminated from the brain, which could reduce the duration of efficacy. This raises a feasibility/translational issue and should be discussed.

Response:

The problem of short-term efficacy of the combinations has been discussed at the end of our Discussion, as suggested

  1. Toxicity: Side effects were a concern for ASMs but not mentioned for borneol and scoparone. The same study by Kowalczyk et al 2022 reported that low nanomolar concentrations of scoparone in the brain were sufficient to induce anxiogenic effect in mice, and that combination of scoparone (12.5 mg/kg) and borneol not only significantly increased brain scoparone concentration but also the level of the toxic compound isofraxidin. Doses more than 200 mg/kg were needed to achieve ≥50% protection from MES (Fig 2). In two-drug combination assays, 50 mg/kg dose was sometimes needed to achieve significant efficacy (Table 1), while 25 mg/kg was used to achieve efficacy in three-drug combination (Table 3). Authors should discuss the potential toxicity of using such high doses and how that might impact feasibility.

Response:

Following both Reviewers’ suggestions, we have added a part of the manuscript describing the problem of evaluating the side effects when testing naturally occurring substances, as suggested

  1. Reference #41 (Łuszczki, J.J.; Czuczwar, S.J. 2003) cited in section 4.5 (line 451-458) did not report about fluorescence polarization immunoassay method or preparation of brain tissue/measurement of ASM content. Please amend reference and describe the details in methods including how and when the brains were collected.

Response:

The reference has been changed and we have described this procedure in more detail.

  1. In table 2, it was assumed that the number 50 in bracket represents the dose. If so, please state in the table legend.

Response:

We added information about the doses of the compounds used in this study to the legend of Table 2 and Table 3, as recommended.

Reviewer 2 Report

In this manuscript, the authors assess the anticonvulsant effects of scoparone (a coumarin) and borneol (a bicyclic monoterpenoid) when administered separately and in combination, together with their effect on the antiseizure effects of carbamazepine, valproate, phenytoin, and phenobarbital using the MES model. Assessments for the two-drug and three-drug mixtures were performed using isobolographic transformation of data, while polygonograms were used to show the types of interactions taking place among the selected drugs. The authors illustrated in adequate detail that both scoparone and borneol resulted in seizure attenuation in the MES mouse model, and showed synergistic interactions with some of the classic ASMs. This is an interesting, timely, and well-planned study and it was a pleasure reviewing it. I only have some minor comments that could be taken account to improve the manuscript.

Comments:

Abstract:

-          Please remove from line 16: ‘a serious neurological disease’

Introduction:

-          Can the authors explain in more detail on how coumarins potentiate certain ASMs?

-          What type of side effects do scoparone and borneol have? Is their anticonvulsant mechanism of action understood?

Results/Conclusion:

-          Can the authors detail more on the translatability to the clinic? This could be done in various ways, such as detailing more on what the MES mouse model represents (what type of seizures clinically) etc.

Author Response

REV_2

In this manuscript, the authors assess the anticonvulsant effects of scoparone (a coumarin) and borneol (a bicyclic monoterpenoid) when administered separately and in combination, together with their effect on the antiseizure effects of carbamazepine, valproate, phenytoin, and phenobarbital using the MES model. Assessments for the two-drug and three-drug mixtures were performed using isobolographic transformation of data, while polygonograms were used to show the types of interactions taking place among the selected drugs. The authors illustrated in adequate detail that both scoparone and borneol resulted in seizure attenuation in the MES mouse model, and showed synergistic interactions with some of the classic ASMs. This is an interesting, timely, and well-planned study and it was a pleasure reviewing it. I only have some minor comments that could be taken account to improve the manuscript.

Comments:

Abstract:

-          Please remove from line 16: ‘a serious neurological disease’

Response:

This unnecessary expression has been removed from the Abstract

Introduction:

-          Can the authors explain in more detail on how coumarins potentiate certain ASMs?

Response:

Potentiation of the ASMs by coumarins has been explained in the Introduction, as suggested.

-          What type of side effects do scoparone and borneol have? Is their anticonvulsant mechanism of action understood?

Response:

We have added information about the acute adverse effects that might be produced by active naturally-occurring compounds that can be detected during the evaluation of the anticonvulsant properties of the studied drugs in animals.

Molecular mechanisms of the anticonvulsant action of scoparone are not confirmed as yet, but considering the fact that scoparone is closely (structurally) related to imperatorin, molecular mechanisms described for the latter coumarin can be similar as for scoparone. Description of the molecular mechanisms of action of coumarins are presented elsewhere: Skalicka-Woźniak K, Orhan IE, Cordell GA, Nabavi SM, Budzyńska B. Implication of coumarins towards central nervous system disorders. Pharmacol Res. 2016 Jan;103:188-203. doi: 10.1016/j.phrs.2015.11.023.

Results/Conclusion:

-          Can the authors detail more on the translatability to the clinic? This could be done in various ways, such as detailing more on what the MES mouse model represents (what type of seizures clinically) etc.

Response:

Information about the MES model has already been presented in the Introduction, but, we have focused more on the translatability of the results in the Discussion, as recommended by both Reviewers.